# Understanding what happens to attendees after an NHS Health Check: a realist review

Claire Duddy [ORCID],[1] Erica Gadsby [ORCID],[2] Vivienne Hibberd,[3] Janet Krska,[4] Geoff Wong[5]

¹Nuffield Department of Primary Care Health Sciences, University of Oxford, Oxford, UK
²Faculty of Health Sciences and Sport, University of Stirling, Stirling, UK
³Public Involvement in Pharmacy Studies Group, University of Greenwich Medway School of Pharmacy, Chatham, UK
⁴Medway School of Pharmacy, Universities of Greenwich and Kent, Chatham, UK
⁵Nuffield Department of Primary Care Health Sciences, Oxford University, Oxford, UK

**Correspondence to**
Claire Duddy;
claire.duddy@phc.ox.ac.uk

## ABSTRACT

**Objectives** The NHS Health Check offers adults aged 40–74 an assessment of their risk of developing cardiovascular disease. Attendees should be offered appropriate clinical or behavioural interventions to help them to manage or reduce these risks. This project focused on understanding variation in the advice and support offered to Health Check attendees.

**Design** We conducted a realist review, assembling a diverse body of literature via database searches (MEDLINE, Embase, CINAHL, HMIC, Web of Science) and other search methods, and synthesised data extracted from documents using a realist logic of analysis. Our aim was to develop an understanding of contexts affecting delivery of the NHS Health Check and the underlying mechanisms producing outcomes related to the offer for attendees post-Check.

**Results** Our findings demonstrate differences in how NHS Health Check commissioners, providers and attendees understand the primary purpose of the programme. A focus on screening for disease can produce an emphasis on high-volume delivery in primary care. When delivery models are organised around behavioural approaches to risk reduction, more emphasis is placed on advice, and referrals to 'lifestyle services'. However, constrained funding and competing priorities for providers limit what can be delivered within the programme's remit. Attendees' experiences and responses to the programme are affected by how the programme is delivered, and by the difficulty of incorporating its outputs into their lives.

**Conclusions** The remit of the NHS Health Check should be reviewed with consideration of what can be effectively delivered within existing resources. Variation in delivery may be appropriate to meet local needs, but differences in how the programme's primary purpose is understood contribute to a 'postcode lottery' in post-Check advice and support. Our findings underline existing concerns that the programme may generate inequitable outcomes and raise questions about whether it can deliver positive outcomes for the majority of attendees.

**Trial registration number** PROSPERO CRD42020163822

## BACKGROUND

The NHS Health Check in England is a large-scale public health programme that aims to offer adults aged 40–74 a 5-yearly assessment of their risk of developing cardiovascular disease (CVD) (excluding those with a pre-existing

---

## STRENGTHS AND LIMITATIONS OF THIS STUDY

⇒ This is the first realist review focused on the NHS Health Check programme and it has generated new understanding of the causes of variation in delivery at the end of the Health Check pathway.
⇒ The review is inclusive of a wide range of literature, including grey literature, allowing us to draw on learning from local delivery models.
⇒ This project was strengthened by strong and consistent patient and public involvement and professional stakeholder input, which shaped and tested the relevance of our theories throughout.
⇒ As with any review, our findings were limited by the availability and quality of the available evidence—we sought to fill a gap in the existing literature focused on the end of the Health Check pathway but this necessarily limited the number and quality of documents available.

---

CVD-related condition).[1] The NHS Health Check involves the measurement of multiple CVD risk factors and the delivery of advice and discussion of appropriate clinical and behavioural approaches that might help individuals to manage and reduce their CVD risk. These could include, for example, referral to a General Practitioner (GP) to discuss recommended pharmacological options (generally statins or antihypertensives) or the delivery of advice, signposting or formal referral to 'lifestyle' services, such as smoking cessation and weight management programmes.[2]

The programme is concerned with both the early identification of risk factors for CVD and the delivery of early interventions to address them, via both pharmacological and non-pharmacological, 'lifestyle' or 'behavioural' means. It is also tasked with helping to address population-level health disparities, reflecting an understanding that the burden of CVD is not spread equally throughout the population, but is strongly linked to factors including deprivation and ethnicity. The intention to use the programme to address

health inequalities has been present since its launch; most recently, the programme is cited in the UK Government's recent 'Levelling Up' white paper as a potential means of delivering its prevention agenda.[3] To meet its aims of preventing CVD at a population level, and contributing to reducing the health inequalities associated with CVD, the NHS Health Check must be widely taken up (especially by those with most to gain), and be effective in both assessing and reducing or managing any risks identified.

The NHS Health Check was first introduced in 2009 but the programme was relaunched in 2013, when responsibility for commissioning the programme transferred from NHS Primary Care Trusts to local authorities (LAs). Minimum standards for programme delivery—focused on the collection of certain measurements and targets to increase the volume of checks delivered—became statutory requirements at this time[4 5] and Public Health England (PHE) was formed and given oversight of the programme. In 2021, the new Office for Health Improvement and Disparities (OHID) took on responsibility for the programme and published a major review (instigated and conducted by PHE), setting out an updated 'vision' for the NHS Health Check. The review's recommendations include aims to launch a digital service and to increase the size and remit of the programme by extending it to cover younger age groups and to address more conditions beyond CVD.[6]

The NHS Health Check has long been the subject of controversy, with some calling its effectiveness and cost-effectiveness into question,[7–10] and PHE responding by collating evidence that supports the programme under the auspices of the Expert Scientific and Clinical Advisory Panel.[11] Observational studies included in two PHE-commissioned rapid reviews suggest that the programme is associated with increased rates of CVD risk factor and disease detection, statin prescribing and referrals to 'lifestyle services' (including smoking cessation, weight management, exercise and alcohol support services). Work undertaken to support the recent PHE/OHID review of the programme has also identified an association between Health Check attendance and improvement in a range of indicators, including body mass index (BMI), rates of smoking, blood pressure, total cholesterol, hospital admissions related to CVD or type 2 diabetes and all-cause death, after 5 years,[12] although the direction of causality is unclear.

Regional and local studies of the implementation and outcomes of the programme demonstrate wide variation in programme delivery and outcomes across England. Evidence on the delivery of advice, referrals and on behaviour changes post-Health Check is especially sparse. The rapid reviews identified only six primary studies examining behaviour change, all focused only on smoking cessation.[13 14]

Our scoping searches and an initial review of the existing research evidence relating to the programme identified a clear focus on invitation, uptake and coverage of the NHS Health Check. Far less attention has been paid to what happens after the measurements and risk assessments have been undertaken, especially in relation to the delivery of advice, onward signposting or referral and ongoing support for behaviour changes that might reduce CVD risks. The capacity of the NHS Health Check to provide attendees with appropriate advice, referrals and support—and the extent to which attendees respond to these—is a critical assumption underpinning the effectiveness of the programme.

Current best practice guidance makes recommendations for NHS Health Check commissioners and providers, describing a range of possible advice and referral options that may be made available, as well as clinical interventions that may be appropriate to offer attendees.[2] It is clear that responsibility for clinical follow-up rests with primary care, and the guidance suggests that commissioners 'may wish to' put referral pathways to onward services, such as smoking cessation services, in place. Patel *et al*'s large-scale observational study of the NHS Health Check for the period 2012–2017 includes data relating to the provision of 'advice, information or referral' for different risk factors that may be flagged during a check.[15] Although data recording for these activities is likely to be incomplete, the figures presented in this study suggest that there is wide variation in the delivery and recording of these activities for different risk factors, and that rates of delivery fall well below the recommended thresholds for intervention. For example, only one quarter of attendees recorded as meeting the threshold for intervention are recorded as receiving advice, information or referral in relation to diet. In addition, this national-level study obscures the wide regional variation in what they term 'postdelivery management' following a check, identified in existing reviews that include local studies.[13 14]

In consultation with our stakeholder groups (see Methods ection below for more details on group membership) we focused our review on these final steps in the NHS Health Check pathway. Our aim was to develop an understanding of how the NHS Health Check programme works in different settings, and for different groups to achieve its outcomes, with a specific focus on what happens after the measurements and risk assessment are complete.

## METHODS

We conducted a realist review to synthesise evidence that could help us to develop an understanding of the important contexts that influence the delivery of the NHS Health Check and the mechanisms that produce intended and unintended outcomes. Realist review is an interpretive, theory-driven approach, chosen because existing research clearly demonstrates that the NHS Health Check programme is a complex intervention with context-sensitive outcomes. Our realist analysis used data extracted from the literature to develop context–mechanism–outcome configurations (CMOCs). CMOCs are causal explanations that describe why and how (by

which mechanisms) particular outcomes are generated in particular contexts. Following the realist approach, our findings are an interpretation of a constellation of data extracted from the documents, built on direct evidence but also on silences and contrasts observed across multiple data points extracted from multiple documents.

The NHS Health Check programme is characterised by wide variation in commissioning and delivery and hence in the experience of attendees. Our review sought to make sense of this variation, by developing a realist programme theory, based on CMOCs, that elucidates the causes of outcomes related to the final steps in the patient pathway, that is, on what happens after the measurements and risk assessments are complete.

Our methods are described in brief below, while full details are available in a detailed protocol paper.[16] The conduct and reporting of the review followed the RAMESES (Realist and Meta-narrative Evidence Syntheses: Evolving Standards) quality[17] and reporting standards.[18] A glossary of realist terminology and RAMESES checklist are included in online supplemental file 1.

### Patient and public involvement

Our review was informed by the involvement of two stakeholder groups, who provided us with content expertise based on their lived experience of commissioning, providing or receiving (or being eligible to receive) the NHS Health Check. Our patient and public involvement (PPI) group comprised 10 members of the public from six English regions and were selected to be as diverse as possible in relation to age, gender, ethnicity and geography. Our 'professional' stakeholder group were recruited via our existing networks and snowballing, and included policymakers, LA commissioners, NHS Health Check providers and a trainer and representatives from relevant health charities. Our final meeting with the latter group was expanded to include 36 individuals, to ensure a broad range of feedback on our findings and recommendations for policy and practice.

Both groups were consulted via regular online meetings throughout the project. We asked these groups to provide feedback and responses to our emerging findings as the project developed. For example, we asked them to reflect on how our findings fit with (or did not fit with) their own knowledge and experience of commissioning, providing or attending an NHS Health Check. These discussions helped to shape our analysis, highlight areas of importance to different groups and inform the development of our recommendations. Our PPI group consistently observed a 'mismatch' between the focus of the NHS Health Check programme and their own health priorities, and a lack of understanding of the programme's purpose. This informed our analysis of data relating to attendees' understanding of their health needs and our overall focus on different groups' understanding of the purpose of the programme.

In our final meetings with each group, we asked our stakeholders for their input to help us to develop and refine recommendations to inform NHS Health Check policy, commissioning and delivery. We also sought their input to help us to develop appropriate dissemination strategies, tailored project outputs and important audiences to share our findings.

### Realist review methods

Our realist review followed Pawson's five iterative steps,[19] as described in detail in our protocol.[16] These steps are summarised in table 1 below. We used the Preferred Reporting Items for Systematic Reviews and Meta-Analyses (PRISMA) checklist (where applicable) when writing our report (see online supplemental file 1 for details).[20]

As the project progressed, we made some minor changes to the protocol. In Step 2, we anticipated a potential need for further searching for empirical data, but our project team agreed that the processes described above had identified sufficient material to meet our needs. During the analysis, we conducted a series of focused searches to identify material related to one substantive theory—'street-level bureaucracy', which has been used to support our explanations of variation in delivery of the NHS Health Check.[21] The details of these searches are also reproduced in online supplemental file 2.

For Step 3, our original criteria were updated in light of the review project's chosen focus on what happens after measurements and risk assessments are completed. We sought in particular documents that contained data relating to later steps in the NHS Health Check pathway, including the provision of advice, signposting, referrals or prescriptions, and outcomes associated with these.

In Step 4, CD and EG coded a 10% set of the included documents independently and GW provided an additional check to ensure that all relevant data were captured and coding was applied consistently.

## RESULTS
### Documents included in the review

In total, 124 documents contributed data to the review, including 59 published research papers or reports; 20 documents reporting local evaluations; 34 conference materials (presentations, abstracts and posters) and 11 others (including policy reports, guidance, news articles and theses). Of these, 21 contributed data relating to the NHS Health Check programme as a whole, and the rest focused on specific localities. The processes of identifying, screening and selecting these are summarised below in figure 1 (an adapted version of the PRISMA flow diagram).[20]

The main characteristics of the included documents are provided in online supplemental file 3. This table also indicates the extent to which each document contributed to the analysis, by showing the contribution of extracted data from each to specific CMOCs.

**Table 1** Summary of methods

| | Aim | Approach |
|---|---|---|
| Step 1: Locate existing theories | To identify existing theories that offer explanations from when, how and for whom the NHS Health Checks programme 'works'. See online supplemental file 1 | Informal literature searching, focused on existing reviews and programme documentation (CD) Drawing on experience and knowledge of project team and stakeholders (All) |
| Step 2: Search for evidence | To gather a body of literature containing data that could be used to refine the initial programme theory. | Screening material included in two existing reviews[13 14] Additional searches across multiple databases (see online supplemental file 2) Trawling NHS Health Check website for grey literature Citation chaining Email alert to capture new material (CD) |
| Step 3: Article selection | Within the material identified as described above, to select documents that could contribute relevant data for theory development. | Screening title and abstract and then in full text (CD) Inclusion of documents based on assessment of relevance and rigour[41] 10% of records screened in duplicate, with discrepancies resolved by discussion (CD, EG, GW) |
| Step 4: Extracting and organising data | To describe and organise included documents. To extract and code relevant data from included documents and begin analysis. See online supplemental file 3. The full set of extracted data are available on request. | Key characteristics of documents captured in an Excel spreadsheet (CD) Relevant data related to our research question and focus coded in NVivo 10% of records coded in duplicate and checked by a third reviewer (CD, EG, GW) |
| Step 5: Synthesising evidence and drawing conclusions | To apply a realist logic of analysis to extracted data to develop CMOCs related to the end of the NHS Health Check pathway. CMOCs are the building blocks of realist analysis, defining relationships between important outcomes, the mechanisms that produce them and the conditions in which they are likely to occur. To develop a final programme theory. | Close examination and interpretation of data coded within each category to build CMOCs (CD) Cross-case comparison to identify data that demonstrated similar or contrasting contexts and mechanisms that produced patterns of outcomes. Iterative development of CMOCs as more data were considered and discussed by the project team and stakeholder groups (All) |

CMOCs, context-mechanism-outcome configurations.

## Review findings

Our findings describe what our included data told us about what happens at the end of an NHS Health Check, after the measurements and risk assessments are completed. Our explanations describe the reasoning and responses of three important groups involved in the programme: LA commissioners, NHS Health Check providers and attendees. A narrative summary of our findings follows; it is based on our realist analysis that developed 86 CMOCs underpinned by the data included in the review. Overall, our findings illustrate how the delivery and outcomes of local NHS Health Check programmes and individual Heath Check encounters are shaped by a wide range of influences. To provide transparency, a detailed summary of the CMOCs developed and the data underpinning each are available in online supplemental file 4. We have included references to illustrative CMOCs to support the narrative that follows. CMOCs are labelled according to the perspective that they focus on, that is, commissioner-focused CMOCs are designated 'C', provider-focused CMOCs 'P' and attendee-focused CMOCs 'A'.

### Understanding and engagement with the NHS Health Check

Our review findings point to variation in understandings of the primary purpose of the NHS Health Check. This variation is a critical factor that determines how the programme is commissioned, delivered and received. The programme was designed with two aims in mind: early identification of cases of diagnoseable illness or individuals with 'high' risk, and early intervention to support individuals to manage and reduce their CVD risk, via prescribing or by prompting or enabling behaviour change. Further to these, the programme is also cast as an opportunity to address disparities in CVD risk factors across England, although it maintains a focus on individual behaviours. While there is a focus in the literature on inequality in relation to invitation and coverage of the check,[22 23] our review uncovered very little data and almost no focus on the relationship between inequalities and the delivery of advice and referral. This gap is also reflected in the evidence collated to support the recently published PHE/OHID-led review.[12]

Our analyses of the data suggest that commissioners and providers may have a tendency to prioritise some

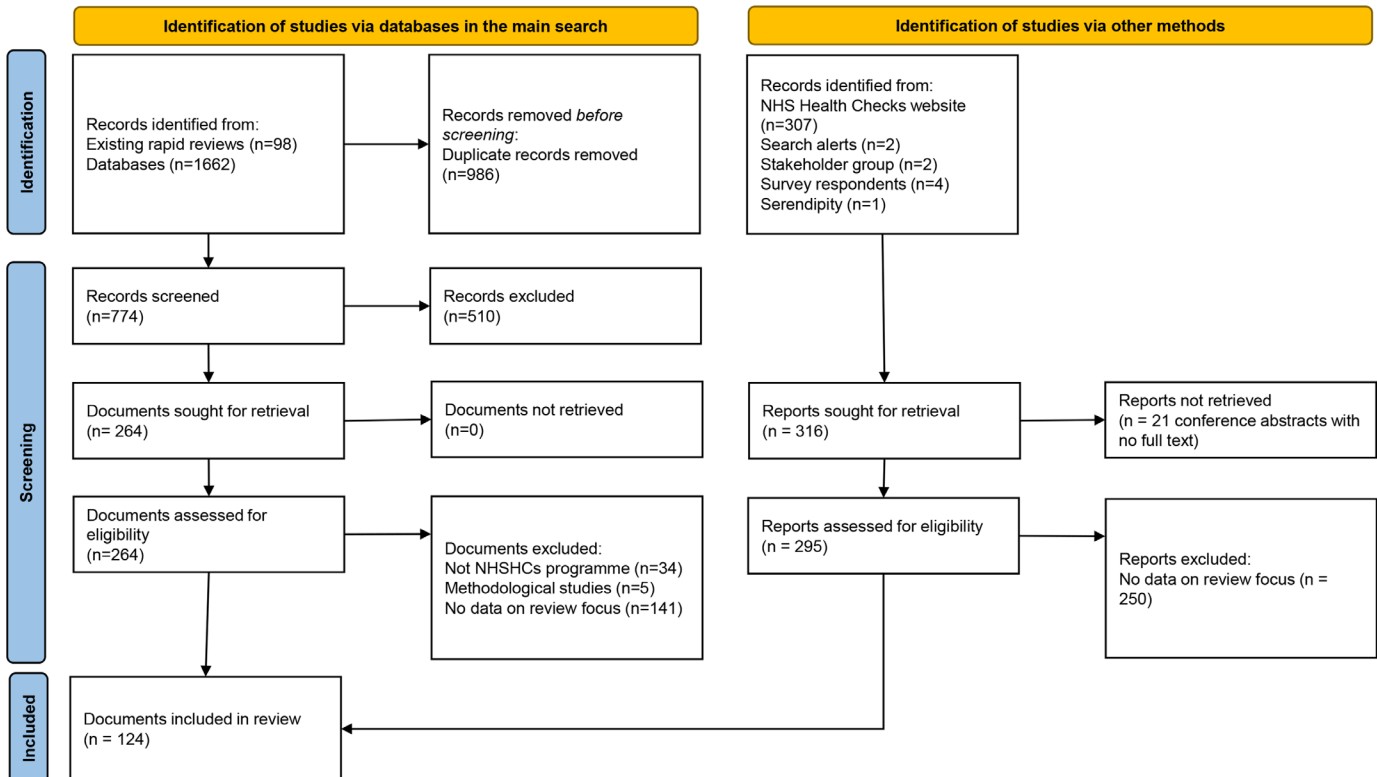

**Figure 1** Summary of searching and selection processes.

aspects of the programme over others, for example, increasing volume of delivery and delivery of risk assessments over risk management. This prioritisation is then reflected in local commissioning decisions and delivery models. When the programme is understood primarily as an opportunity to screen for cases of existing CVD or near-CVD (very high risk factors), responsibility for programme delivery and outcomes is likely to rest with primary care, and specifically in general practice, where most NHS Health Checks are provided, and where clinical and especially pharmacological follow-up should happen (see, eg, *CMOCs C9–C10* in online supplemental file 4). This perspective leads to an emphasis on high-volume throughput, increasing coverage of the programme and a focus on efficient delivery of the minimum requirements for each check, collecting mandatory data and communicating risk scores. At the other end of the spectrum, where commissioners and providers are more oriented towards using the programme as a means of supporting behaviour change in order to reduce or manage CVD risk, delivery reflects this, with greater emphasis on advice or coaching, and facilitating onward referrals to 'lifestyle services', such as smoking cessation or weight management programmes (*CMOCs C1–C8*).

Our data demonstrate how such differences in local priorities for the NHS Health Check can drive variation in who delivers the check and the training they receive, the settings in which checks are delivered and the time allocated to each check (*CMOCs C1–C8, P14, P26*), all of which go on to affect how attendees experience the programme (*CMOCs A12–A17, A30–A37*). In addition,

these differences can affect the availability, accessibility and connectedness of referral pathways to local 'lifestyle services' that might be offered to attendees post-Check, and important differences in the information that LAs collect, how they monitor and evaluate programme delivery, as well as funding models that can incentivise certain aspects of delivery (*CMOCs C2–C3, C5–C8*). Commissioners and providers each influence how the programme is shaped and implemented at local levels, exercising discretion in determining the remit and focus of the programme.

This double layer of discretion in how the NHS Health Check is ultimately enacted in different local areas means that the relationship between LA commissioners and providers is an important factor in determining what local delivery looks like. Commissioners determine service specifications and funding models and the scope of monitoring of programme delivery, but (unless they are themselves providers, as is the case for some provision in a few LAs) their day-to-day influence on actual delivery is limited (*CMOC C4*). This is a particular concern as prevailing scepticism (in particular among GPs) about the programme's effectiveness and concerns about the potential for overdiagnosis has led to disengagement among some providers. Such concerns, however, must be balanced with their reliance on income generated by the programme (*CMOCs P1–P5*). Differences in levels of engagement with the programme can reflect different understandings about its primary purpose and concerns about its ability to meet its aims, but they can also be fuelled by more practical concerns, including competing

priorities and workload pressures, and the need to deliver NHS Health Checks and appropriate follow-up with limited resources (see below and *CMOCs P9–P13, P15–P18*).

Local delivery models send important signals to attendees that inform how they understand and engage with the programme in turn. Checks delivered in general practice with a focus on completing mandatory measurements and risk assessments carry the implication that the check should be understood as a screening test or a health 'MOT', that may or may not result in the need for clinical intervention (*CMOCs A1, A12*). Delivery in community settings, by providers trained specifically to deliver coaching or behavioural support, or otherwise with a greater focus on delivering advice or referrals, sends a different message (*CMOC A14*). In some areas, NHS Health Checks have been delivered within a wider integrated lifestyle service, facilitating connections with lifestyle services, and potentially additional services such as link workers or social prescribers. These models may enable providers to offer more holistic support that better meets the needs of individual attendees; much data from the attendee perspective suggest that this may help to generate better engagement with the programme and its aims.

Our data also suggest that attendees' prior expectations of the check also inform their response, and so it is important that national and local advertising about the programme are clear about its remit and purpose. Providers can transmit 'soft' signals to attendees about the programme's purpose and value, and attendees will pick up on these (*CMOCs A17–24*). They have the potential to convey urgency, and even induce fear and anxiety in attendees, but they may also (intentionally or otherwise) imply that the check is a 'box ticking' exercise.

Finally, it is important to acknowledge that attendees' themselves can exercise discretion in their response to their check, regardless of how providers approach its delivery. The programme's focus on individual behaviour creates high expectations of individual action to address any 'lifestyle risk factors' identified. Inevitably, attendees may face considerable challenges in implementing behaviour changes within their own lives, where they too may have limited resources (*CMOCs A38–A34*). Some literature has drawn attention to the risks inherent in this individual focus, including the potential for positive health impacts to be realised only by those with the capacity to make significant lifestyle changes, and so unfairly distributing any benefits among attendees.[24] Such a focus risks increasing rather than decreasing health inequalities.

### Practical constraints limiting the NHS Health Check

Commissioners and providers face substantial practical constraints that limit and inform their exercise of discretion in programme delivery. As noted above, the decisions of LA commissioners that are enacted in funding models, service specifications and monitoring regimes set important boundaries for providers, potentially incentivising different delivery methods and priorities. Commissioners' decisions must take into account restricted funding for public health initiatives (*CMOCs C12–C14*). Central government grants to LAs have fallen substantially since 2010. Since public health responsibilities were transferred to LAs in 2013, the public health grant has decreased by 13% in real terms.[25] Spending on the NHS Health Check programme fell by 21% between 2015/2016 and 2019/2020, and spending on 'lifestyle services' that could potentially support attendees post-check has also fallen: expenditure on smoking cessation initiatives fell by one third over the same period, adult alcohol and drug services by 17% and weight management services by 5%.[26] Reductions in funding of this scale necessarily limit what commissioners can purchase (*CMOCs C14, A35–A37*). Our findings raise the question of whether public health funding overall is adequate to support the programme's more ambitious aim of preventing CVD and reducing CVD inequalities by reducing or managing behavioural risk factors. In particular, it is apparent that the delivery of personalised, individual advice and discussion during checks, and the offer of further support afterwards, is a more intensive proposition than a programme focused on case finding (*CMOCs P9–P13*).

Providers face other constraints. There is a need to ensure that staff involved in delivering checks are competent to do so, but some data suggest that not all providers feel confident in delivering support for behaviour change (*CMOCs P20–P23*). Training, too, must reflect the programme's aims both to identify and to address CVD risk factors (*CMOC P26*). Wherever checks are delivered in general practice or community pharmacy settings, providers face competing priorities and demands on their time. Workload and time pressure may push providers towards 'leaner' delivery, focused on mandatory tests and capturing required measurements, leaving little time for discussion, advice and offers of referral (*CMOCs P9–P14*).

There is a crucial relationship between the practical constraints that commissioners and providers face and their understanding and engagement with the programme. It is likely that both groups of actors adapt their understanding of the programme based on what they know about the limited resources available to deliver and support it. Downward pressure on funding inevitably incentivises leaner delivery models and, as our findings make clear, these models tend to favour the 'early identification' or case finding purpose of the programme. Provider scepticism is also likely to be compounded by the sense that both the programme itself, and subsequent services on which it might depend are underfunded and inaccessible (*CMOCs P1–P3*).

Finally, our data make clear the impact of the varied delivery models on attendees' experience of the programme and on what they are offered (or not) post-check (*CMOCs A1, A5–A6, A11–A14, A30–A32, A34, A38–A40*). It is unclear which delivery models allow or incentivise providers to deliver the meaningful,

personalised and ongoing advice and support attendees might need. Existing ethnographic research has demonstrated that the time-constrained and highly structured nature of the NHS Health Check assessments impede meaningful discussion that prioritises understanding individuals' circumstances.[27 28] Limited access to 'lifestyle services' means these are inaccessible to many.

## Variation in NHS Health Check delivery models: street-level bureaucracy

Our review findings demonstrate wide variation in how the NHS Health Check programme is implemented locally, with a specific focus in variation in the delivery of what happens after measurements and risk assessments have been completed. Inconsistencies in the recording of these activities (especially the delivery and uptake of advice and referrals post-check) prevent the development of a comprehensive picture of this variation across England. However, our review findings provide a starting point to improve understanding of what influences delivery in these areas, highlighting the discretion available to LA commissioners and individual NHS Health Check providers in making decisions about how the programme is delivered on the ground. In addition, they highlight how differences in delivery models affect how the checks are experienced by attendees, and what colours their responses to the information they receive and any offers of further intervention.

Our understanding of the processes at work in driving variation draws on Lipsky's concept of 'street-level bureaucracy', borrowed from the international relations literature.[21 29] Previous research on the NHS Health Check,[30] and the implementation of other health policies in the UK[31 32] have used the same theory to add explanatory value. Lipsky's framework emphasises the discretion of those charged with implementing national policies or programmes, as well as their responses to working with limited resources. For the NHS Health Check programme, it is clear that while commissioners and providers are working within the broad constraints of a legal framework[4 5] and guidance issued by PHE[2] (now OHID), their decisions and everyday practice in delivering checks effectively determine the remit and purpose of the programme at local levels. As Lipsky describes it: '*the decisions of street-level bureaucrats, the routines they establish, and the devices they invent to cope with uncertainties and work pressures, effectively become the public policies they carry out*'.[29]

The extent to which discretion can be exercised in relation to the NHS Health Check may be greatest at the end of the programme pathway. Processes relating to earlier steps—the identification of the eligible population, invitation and the actual measurements and risk assessments to be administered—are restricted by the programme's legal framework and clear guidance, leaving little room for local interpretation or adaptation. The later steps—especially the delivery of advice and referrals—are less prescribed and more dependent on local delivery models

and other local services. Activity in these areas is less well recorded and monitored, leaving LA commissioners and providers with more discretion to determine what day-to-day local delivery of checks will look like.

Although Lipsky's original framework focuses on those directly engaged in the delivery of policies and their interaction with the recipients or subjects of those policies (in our case, the interaction of providers with attendees), our analysis also highlights the discretion of LA commissioners who make decisions about local programme specifications and support. While they may not directly interact with the public, commissioners must also interpret the requirements of the programme and exercise their own discretion to ensure it meets local needs and is delivered within local constraints. The approach of the LA directly affects commissioned providers, and this double layer of discretion forms the local contexts in which NHS Health Checks are delivered across England.

Taken together, the CMOCs developed in this review point to the influence of both commissioners' and providers' understanding and engagement with the programme, as well as the effects of practical constraints that drive decision-making in relation to programme delivery. These two aspects mirror Lipsky's concepts of *discretion* in the enactment of policies and programmes, and the effects of their responses to *limited resources*. Attendees' experience of, and response to the programme is affected both by the outcomes of commissioner and provider decisions, but also by other external factors related to their individual circumstances, which may be difficult to align with what the NHS Health Check expects of them.

## DISCUSSION
### Summary of findings
The success of the NHS Health Check programme as an intervention that aims to support individuals to manage and reduce their CVD risk rests on what happens to attendees when the measurements and risk assessments have been completed. While case-finding is an important indicator of success, the delivery of advice, offers of signposting, referrals or other support and attendees' responses to these, is equally crucial for the programme to prevent ill-health and reduce inequalities in the longer term. Our focus on this area reflected the importance of this step—confirmed by our two stakeholder groups—but also the relative lack of attention that it has received in the existing research literature. Our review therefore sought to examine the final steps in the NHS Health Check programme pathway, to understand the factors that influence the delivery of these parts of the check and what follows, and how they are received by those who receive checks.

The NHS Health Check as a whole has been the subject of several previous evidence synthesis projects, which have considered many aspects of the programme, and have similarly identified wide variation in delivery

models and outcomes. Our review is the first to use a realist approach, and the first to focus specifically on the steps that follow the measurements and risk assessments made during each check. Relevant findings from existing reviews correspond with our own. The two PHE-commissioned rapid reviews identified some limited evidence of geographical variation in referrals, and captured qualitative data describing providers' doubts and scepticism about the programme, and perceived training needs. These reviews also included qualitative studies focused on attendees, which described attendees' perspectives on the quality of information delivered during checks and the important constraints imposed by 'environmental', 'resource' and 'time' factors that limit attendees' capacity to make and sustain behaviour changes.[13 14] Another synthesis coded evidence relating to the behaviours of commissioners, providers and attendees in relation to the programme, and reported similar findings related to providers' skills, attitudes and beliefs, as well as the resources available to deliver checks and the need to take account attendees' wider 'social contexts' when delivering advice.[33]

Our review extends the work undertaken in these reviews, by explaining how and why variation in how the programme is delivered comes about, with a particular focus on what happens after the measurements and risk assessments are completed. Our findings point to the significance of the exercise of discretion by LA commissioners and NHS Health Check providers in the delivery of checks. Commissioners' decisions in relation to programme implementation, funding and monitoring, and providers' actions to deliver checks on the ground are influenced by multiple factors. These include their understanding of the primary purpose of the programme and engagement with its aims, but these attitudes towards the programme—and commissioners' and providers' actions in organising and providing checks—are constrained by important practical factors.

Prevailing conditions, including limited funding for public health programmes and services overall, current funding and monitoring arrangements for the NHS Health Check programme, and competing priorities for many providers tend to push towards a delivery model that prioritises the programme's aim of early detection—case finding. The prioritisation of this element of the programme may be to the detriment of work that could help to prompt or support behaviour change. These constraints raise ethical questions about the programme: if it is oriented towards early identification of risk, but cannot genuinely support early intervention to help manage and reduce risks (bar prescribing), where does this leave attendees? There is a need to consider the relationship between the NHS Health Check programme and the wider landscape of local and national services that can be offered to attendees.

In some local areas, there is evidence that commissioners and providers work against this tide, using their local NHS Health Checks as a means of supporting attendees to make behaviour changes that could help to reduce their CVD risk. Our final programme theory diagram summarises these findings and is presented below in figure 2.

## COVID-19 and the NHS Health Check

We undertook this review during the COVID-19 pandemic, which had a major impact on delivery of the NHS Health Check. In April 2020, the programme was effectively paused.[34 35] To support resumption of delivery, PHE issued 'restart guidance' in April 2021, encouraging LAs to consider restarting the programme, dependent on local safety assessments and the need to prioritise the vaccination programme.[36] This document acknowledged the pressures faced by general practice in particular, and urged LA commissioners to consider 'alternative' providers. Other communications from PHE and the Department of Health and Social care during the pandemic highlighted the potential benefits of the NHS Health Check in relation to identifying risk factors for severe COVID-19 outcomes, as many CVD risk factors are also associated with higher risks of hospitalisation and death from COVID-19.[37–39]

In light of our findings, we note that the effects of the pandemic have the potential to exacerbate some existing contexts that may adversely affect delivery of the programme's aims. In particular, pressure to 'catch up' and concerns about delayed or missed diagnoses could lead to a more intense focus on early detection and case finding.[40] Workload pressures affecting primary care providers may also increase disengagement from the NHS Health Check, and LA commissioners must consider these effects as they restart the programme in each local area.

## Implications for policy and practice

Our review findings raise a number of important questions for policymakers, commissioners and providers to consider in relation to the NHS Health Check. Our main recommendation is that all three of these groups (and future researchers) should increase their focus on evaluating and improving the delivery of the later steps in the programme pathway, that is, on what happens to attendees after the measurements and risk assessments have been completed. Our more detailed recommendations are summarised below in table 2. They are organised around four important and interconnected principles, developed in consultation with our PPI and professional stakeholder groups: clarifying the purpose of the programme; increasing engagement with the programme; focusing the Health Check on attendee needs; and improving links between the NHS Health Check and other programmes and services.

Our recommendations must be considered alongside those made by PHE/OHID in their recently published

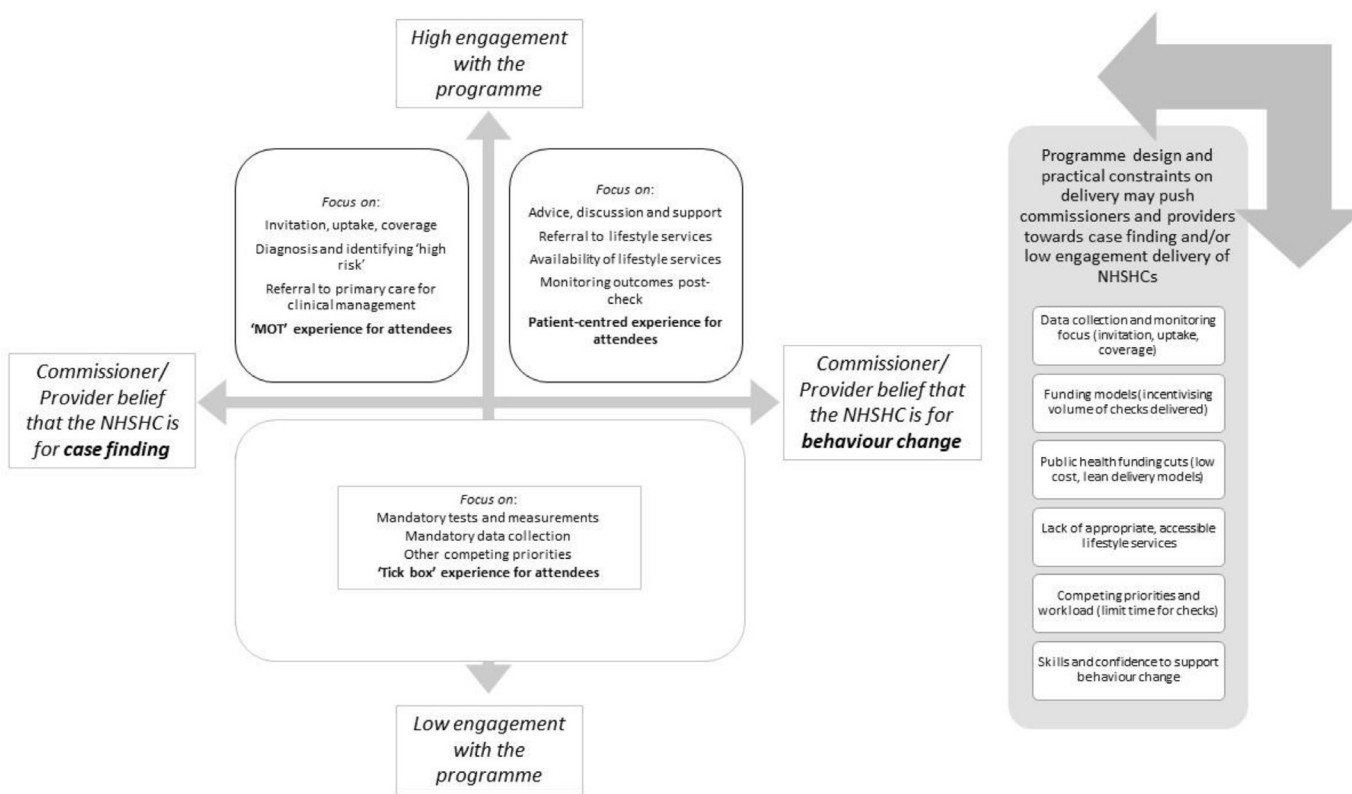

**Figure 2** Final programme theory.

national review of the Health Checks programme.[6] They support several of the recommendations made by OHID, and in particular the identification of the need to 'build sustained engagement' and 'create a learning system'. In

these areas, OHID's specific proposals include ensuring a clear focus on the programme's aim to '(promote) lasting health and well-being'; developing a national training offer with a focus on supporting behaviour change;

| **Table 2** Summary of recommendations based on review findings | |
|---|---|
| **Our recommendation** | **Rationale** |
| Ensure all national and local guidance and programme documentation reflects the importance of the programme's aims in relation to both early identification of CVD risk, and early intervention to manage/reduce that risk. | Different interpretations of the primary purpose of the programme drive variation in commissioning and delivery, and subsequently in attendee experience. |
| Assess overall programme funding and local funding arrangements in relation to delivery of (all aspects of) the programme's aims. | Commissioner, provider and attendee scepticism about the programme can undermine its delivery; inadequate resources for the programme and support services for attendees post-check can reduce engagement. Funding arrangements may incentivise leaner or more intensive checks. |
| Review national and local data monitoring and evaluation of the programme, to ensure that data relating to all key aspects of the programme (and especially the final steps in the NHS Health Check pathway) are captured. | Data collection is currently focused on invitation, uptake and coverage, incentivising high-volume, less intensive delivery that focuses on mandatory data points. Data on advice and referral (including data relating to potential disparities in provision) are needed to evaluate the delivery of these steps. |
| Review training for providers to assess its ability to support all aspects of programme delivery, including the final steps in the NHS Health Check pathway. | Providers and attendees have identified deficiencies in skills and knowledge, especially in relation to the delivery of advice, support and referrals. All providers should receive adequate training to ensure that they can deliver these aspects of a check or signpost/refer attendees to an appropriate source of support. |
| Review connections between the programme and national and local services that could offer further support for attendees, and options for longer term follow-up after measurements and risk assessments are completed. | The success of the programme rests in part on its connections with other services that could offer further support for attendees. Strengthening these connections could allow the NHS Health Check to act as a gateway to these services (as well as to primary care/general practice when necessary). |
| Continue to produce evidence for programme effectiveness and address the relative scarcity of evidence focused on later steps in the NHS Health Check pathway. | Commissioner, provider and attendee scepticism about the programme can undermine its delivery; evidence from research and evaluation should go beyond a focus on increasing coverage of the programme and inform practical recommendations for good practice in delivery. |
| CVD, cardiovascular disease. | |

providing sufficient provision of post-check services; and continued evaluation of the programme.

However, our findings raise concerns about the implementation of some of the recommendations from the national review. In particular, two major recommendations ('start younger' and 'address more conditions') propose to significantly expand the scope and coverage of the NHS Health Check programme. Our findings relating to attendee experience suggest that moves to make the programme more holistic may be welcomed by some, but indicate that policymakers should be cautious about any programme expansion. There is a risk that expanding the programme's scope could add to existing confusion about its primary purpose and drive further local variation in delivery. Existing capacity in primary care and community and public health services also limits the feasibility of these proposals. Without sufficient appropriate follow-up, extensions of the programme's remit risk leaving more attendees with few options for ongoing support to help them to address or manage any risks or conditions identified. Without additional investment in services and convincing evidence of their clinical and cost-effectiveness, such expansion also risks increasing provider scepticism and disengagement from the programme.

OHID's proposal to launch a digital version of the NHS Health Check should also consider our findings, and in particular, ensure that the final steps in the programme pathway are not neglected. In any delivery format, there is a need to ensure that where risks are identified, relevant advice and links to appropriate services are provided. Finally, we note that OHID's recommendation to continue to increase participation reinforces the existing focus on invitation and uptake of the NHS Health Check. Policymakers should be cognisant that measures that encourage and incentivise high-volume delivery of checks can detract from the delivery of high-quality, personalised advice and ongoing support for longer-term behaviour change.

### Strengths and limitations of this review

Our review has developed novel interpretations of existing secondary data relating to what happens at the end of the NHS Health Check pathway. Our close examination of what happens after the measurements and risk assessments have been completed during checks helps to address the relative lack of research on this particular aspect of programme delivery. The review project was strengthened by close working with our diverse PPI and professional stakeholder groups, who helped us to focus the review, provided detailed feedback on emerging findings and shaped our interpretation of the data and the development of recommendations. We included a diverse range of material in the review, and in particular, drew on the learning captured in a wide range of grey literature including local evaluations and conference materials.

As with any review, our findings are limited by the availability and quality of the literature. The material included in the review covers a wide data range and some older material may be less applicable to the present day. However, we considered each piece of data carefully before inclusion and aimed to select material that spoke to still-relevant contexts, mechanisms and outcomes. Our stakeholders also helped us to confirm the contemporary relevance of our findings. Our CMOCs vary in terms of the volume and rigour of the data that underpin them. We have provided a full and transparent account of that data in online supplemental files 3 and 4, so that the strength of each is made clear to readers, such that they can make their own judgement on the plausibility of our interpretations.

## CONCLUSIONS

Our review has revealed wide variation in the delivery of advice and support, and onward signposting and referral for attendees and identified explanations for this variation. We have identified a wide range of influences that affect how LA commissioners and NHS Health Check providers develop and deliver the programme at local levels across England, which affect how attendees experience and respond to their check. Our analysis explains how differences in understanding of the primary purpose of the programme influences commissioning and implementation, and how practical constraints limit what can be delivered within the programme's remit and existing resources. Based on our findings, we developed a set of recommendations for policymakers, commissioners and providers to inform future programme development. Our recommendations centre on the need for greater emphasis to be placed on the final steps in the NHS Health Check programme pathway, including in national and local guidance and programme documentation, funding models, provider training, monitoring and evaluation of the programme, and in considering how the NHS Health Check can be better linked with wider services and programmes.

**Acknowledgements** We would like to express our thanks for the help, feedback and advice provided by all of our wonderful stakeholders and patient and public involvement contributors, and for the continued engagement of PHE and OHID and their help with the recruitment of stakeholders and opportunities for dissemination of our findings and outputs.

**Contributors** JK and GW conceived the research project and all of the authors contributed to its design. GW provided methodological support. CD devised and carried out the literature searches and led the analysis and interpretation of the data. The other authors (EG, VH, JK, GW) contributed to the analysis. VH led the PPI involvement. CD drafted the manuscript and the other authors (EG, VH, JK, GW) provided feedback. All of the authors read and approved the final manuscript. CD is the guarantor.

**Funding** This project was funded by the National Institute for Health Research (NIHR) Health Services and Delivery Research programme (NIHR129209). The views expressed are those of the authors and not necessarily those of the NIHR or the Department of Health and Social Care.

**Competing interests** CD and GW are both members of the Royal College of General Practitioners (UK) Overdiagnosis and Overtreatment Group. GW is an NHS General Practitioner and deputy chair of the National Institute for Health Research Health Technology Assessment Prioritisation Committee: Integrated Community Health and Social Care Panel (A) and member of the Methods Group (A).

**Patient and public involvement** Patients and/or the public were involved in the design, or conduct, or reporting or dissemination plans of this research. Refer to the Methods section for further details.

**Patient consent for publication** Not applicable.

**Ethics approval** Not applicable.

**Provenance and peer review** Not commissioned; externally peer reviewed.

**Data availability statement** Data are available upon reasonable request. The datasets generated and/or analysed during the current study are not publicly available due to copyright and licensing restrictions but are available from the corresponding author on reasonable request.

**ORCID iDs**
Claire Duddy http://orcid.org/0000-0002-7083-6589
Erica Gadsby http://orcid.org/0000-0002-4151-5911

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
