## [Reviewer comments · BMJ Open]

ARTICLE DETAILS

TITLE (PROVISIONAL)	Understanding what happens to attendees after an NHS Health Check: a realist review
AUTHORS	Duddy, Claire; Gadsby, Erica; Hibberd, Vivienne; Krska, Janet; Wong, Geoff

VERSION 1 – REVIEW

REVIEWER	Riley, V Staffordshire University, School of Life Sciences and Education
REVIEW RETURNED	09-Jun-2022

GENERAL COMMENTS	Thank you for inviting me to review this important piece of work. While a lot of work exists around variation in delivery and understanding of the NHS Health Check programme, there is less so that focuses on commissioners and providers and what happens following the completion of an NHS Health Check. I have a few minor comments below: 1. Page 3 (line 49-50) - you describe the programme as available to anyone aged 40-74 which is not strictly accurate. Please review the programme's eligibility criteria and amend accordingly.2. Page 10 - While a glossary is provided in supplementary materials and abbreviations are noted at the end of the review, it would be helpful if 'CMOC's' were defined and explained prior to the the results section. This would make it easier and clearer for the reader to follow the findings and evidence.3. Page 19 - Implication for Policy and Practice - You provide a very helpful table that summarises your recommendations for the programme. It would be useful if you considered your recommendations in light of those already put forward by the recently published national review. You note the review in the introduction but do not revisit this in the discussion. Are your recommendations considered in the national review or are they overlooked?
---

REVIEWER	Bonner, Carissa Western Sydney University School of Medicine
REVIEW RETURNED	27-Jul-2022

GENERAL COMMENTS	This is an interesting review and realist analysis explaining variation in Health Check delivery in the UK. It asks an important question with stakeholder input, and uses a rigorous method in line with published protocol. Variation in stakeholder understanding of the primary purpose of the programme is a particularly important finding affecting delivery. It would be helpful to see a clearer link between the document data and the findings as I can't tell from the
---

	main paper where ideas came from – providing a shorter version of your supp 4 table (e.g. overarching themes, CMOC numbers, documents contributing to this) with document quotes in text would address this. Sections with references to external literature and theory may be better in the discussion. INTRO/METHODS PPI/PRISMA/MOT – explain abbreviations at first instance RAMESES and PRISMA both mentioned but only PRISMA provided in supp file, can you explain RAMESES and how it was used? Method – this is very comprehensive and clear; could explain background of authors and involvement of steps (as per step 4 modification where author initials are first used) RESULTS Much of the detail for results is in supplement 4 but is not clearly connected to the results section. Could you add the overarching themes from supp 4 to the results as the basic structure or in a smaller table without so many CMIC details as there are so many, e.g. “Understanding and engagement with the NHSHC programme: case finding or enabling behaviour change?” Explain what you mean by P, A, C in CMOC references It’s not entirely clear how the documents identified led to the results reported; I’d perhaps include document quotes in main paper to show examples of the data. For example I can’t tell where a statement like this comes from “This perspective leads to an emphasis on high volume throughput” – were documents that described the health checks as a high risk screening programme reporting quantitative outcomes more than others? Bringing details of the CMOCs supp 4 table into the paper would help this, perhaps in this sort of format: (reflected in CMOC 9 from 8 documents, e.g. “[quote from document]”) Where does this finding come from? “This is a particular concern as prevailing scepticism (in particular amongst GPs) about the programme’s effectiveness and concerns about the potential for overdiagnosis has led to disengagement amongst some providers” References to other literature are probably better in the introduction or discussion as it is not clear whether this is from your data when integrated into results: “Central government grants to LAs have fallen substantially since 2010. Since public health responsibilities were transferred to LAs in 2013, the public health grant has decreased by 13% in real terms.(30) Spending on the NHS Health Check programme fell by 21% between 2015/16 and 2019/20, and spending on ‘lifestyle services’ that could potentially support attendees post-check has also fallen: expenditure on smoking cessation initiatives fell by one third over the same period, adult alcohol and drug services by 17% and weight management services by 5%.(31) “ and “rained and highly structured nature of the NHS Health Check assessments impede meaningful discussion that prioritises understanding individuals’ circumstances.(25, 26) “ The street level beaurocracy section of the results doesn’t appear to be linked to your data very closely and is less useful to me – add details or move shorter version to discussion DISCUSSION Table 2 – should this include something on equity? Final theory figure is useful
--	---

VERSION 1 – AUTHOR RESPONSE

Response to reviewer 1

1. COMMENT: Page 3 (line 49-50) - you describe the programme as available to anyone aged 40-74 which is not strictly accurate. Please review the programme's eligibility criteria and amend accordingly.

RESPONSE: Thank you for pointing out this important detail. We have amended the text to make it clear that there are eligibility criteria for the Health Checks programme and added a reference pointing readers to the relevant webpage for more details.
2. COMMENT: Page 10 - While a glossary is provided in supplementary materials and abbreviations are noted at the end of the review, it would be helpful if 'CMOC's' were defined and explained prior to the results section. This would make it easier and clearer for the reader to follow the findings and evidence.

RESPONSE: Thank you for this suggestion – we are aware that we have provided a succinct summary of our methods in this manuscript (following the detailed protocol which was published at the outset of the project). A brief definition of CMOCs is provided on page 6 in the introduction to the methods section, and on page 8 in Table 1 (which summarises the methods). We have also now added additional text to this table (for Step 5) to expand on this.
3. COMMENT: Page 19 - Implication for Policy and Practice - You provide a very helpful table that summarises your recommendations for the programme. It would be useful if you considered your recommendations in light of those already put forward by the recently published national review. You note the review in the introduction but do not revisit this in the discussion. Are your recommendations considered in the national review or are they overlooked?

RESPONSE: Thank you for this observation – our review and the PHE/OHID national review of the Health Checks programme were conducted in parallel and we had developed our recommendations before the national review findings were published. However, we have since considered our findings in light of these and we have inserted some text reflecting on areas of similarity and difference, and potential implications for the recommendations made in the national review from our own work (on page 20).

Response to reviewer 2

1. COMMENT: PPI/PRISMA/MOT – explain abbreviations at first instance

RESPONSE: We have now spelled out all abbreviations in the first instance – many thanks for pointing this out.
2. COMMENT: RAMESES and PRISMA both mentioned but only PRISMA provided in supp file, can you explain RAMESES and how it was used?

RESPONSE: The PRISMA checklist is provided as a supplementary file as this is a publication requirement from the journal for all systematic reviews. The RAMESES quality and reporting standards guide the conduct and reporting of realist reviews. We have explained this on page 7 and we have now added a completed RAMESES checklist to Supplementary File 1 (following the PRISMA checklist) and signposted this in the manuscript (page 7).
3. COMMENT: Method – this is very comprehensive and clear; could explain background of authors and involvement of steps (as per step 4 modification where author initials are first used)

RESPONSE: Thank you for pointing out this omission – we have added relevant author initials to Table 1 (page 8) to clarify who was involved in each step without the need for readers to refer to our protocol. This information is also reiterated in the 'Author contributions' section.

4. COMMENT: Much of the detail for results is in supplement 4 but is not clearly connected to the results section. Could you add the overarching themes from supp 4 to the results as the basic structure or in a smaller table without so many CMOC details as there are so many, e.g. “Understanding and engagement with the NHSHC programme: case finding or enabling behaviour change?”

RESPONSE: Thank you for these comments in relation to making clear the links between our results/findings and the data that underpin them. We have used supplementary files to contain the details of the extracted data due to the volume of data involved in this review, and instead presented the findings in the manuscript itself as a more legible narrative. We have followed the example of other published reviews (1, 2) in adopting a system of effectively ‘citing’ the relevant CMOCs that underpin each aspect of our findings – our aim here was to achieve a balance between readability and transparency, allowing readers to cross-reference the narrative with the details in the supplementary files.

5. COMMENT: Explain what you mean by P, A, C in CMOC references

It’s not entirely clear how the documents identified led to the results reported; I’d perhaps include document quotes in main paper to show examples of the data. For example I can’t tell where a statement like this comes from “This perspective leads to an emphasis on high volume throughput” – were documents that described the health checks as a high risk screening programme reporting quantitative outcomes more than others? Bringing details of the CMOCs supp 4 table into the paper would help this, perhaps in this sort of format: (reflected in CMOC 9 from 8 documents, e.g. “[quote from document]”)

Where does this finding come from? “This is a particular concern as prevailing scepticism (in particular amongst GPs) about the programme’s effectiveness and concerns about the potential for overdiagnosis has led to disengagement amongst some providers”

RESPONSE: To help improve legibility, we have added brief text to explain the ‘C’, ‘P’, ‘A’ labelling of the CMOCs which relate to the different perspectives of commissioners, providers and attendees (page 10). The structure of the narrative weaves together CMOCs from different sections of the tables presented in Supplementary File 4, drawing together related findings from the different perspectives – it is not possible to adopt the summary headings from the Supplementary File in the narrative itself. It is also not possible to provide single quotations of the data to support each aspect of the findings. Following the realist approach, our findings are an interpretation of a constellation of data extracted from the documents, built not only on direct literal quotes, but on silences and contrasts observed across multiple data points extracted from multiple documents. For example, our finding relating to a potential focus on clinical activity and high volume delivery of Health Checks is based on our interpretation of data that describe delivery models with this focus, as well as others that adopt a different stance. Taken together, we have surmised this is a reasonable interpretation of what is going on in some local areas, but there is no single piece of data explicitly demonstrating this. By providing the tables in Supplementary File 4 (and the corresponding CMOC information in the study table in Supplementary File 3) we have aimed for a level of transparency that permits access to the relevant data, such that interested readers could make their own judgement on the plausibility of our claims. We can provide the full file of extracted data on request, but are unable to include it in the supplementary files due to copyright and licencing restrictions.

6. COMMENT: References to other literature are probably better in the introduction or discussion as it is not clear whether this is from your data when integrated into results: “Central government grants to LAs have fallen substantially since 2010. Since public health responsibilities were transferred to LAs in 2013, the public health grant has decreased by 13% in real terms.(30) Spending on the NHS Health Check programme fell by 21% between 2015/16 and 2019/20, and spending on ‘lifestyle services’ that could potentially support attendees post-check has also fallen: expenditure on smoking cessation initiatives fell by one third over the same period, adult alcohol and drug services by 17% and weight management services by 5%.(31) “ and “rained and highly structured nature of the NHS Health Check assessments impede meaningful discussion that prioritises understanding individuals’

circumstances.(25, 26)“

RESPONSE: Thank you for your close reading and picking up on some aspects of the narrative that set our findings in wider contexts (e.g. in relation to funding). We included some specific statistics here (page 13-14) to help the reader better understand the relevant backdrop for these specific findings and to make the findings relating to funding constraints more concrete. We have therefore included direct citations in addition to the ‘citations’ for relevant CMOCs. (We note however that there was an error in the citations themselves, which were missing from the reference list – these now appear as numbers 26 and 27). Thank you also for your observation in relation to the finding relating to the structured nature of the Health Check – the direct citation here was added because there is a specific reference to two particular studies. However we have now also added bracketed ‘citations’ to the relevant CMOCs, for consistency with the rest of the narrative.

7. COMMENT: The street level bureaucracy section of the results doesn’t appear to be linked to your data very closely and is less useful to me – add details or move shorter version to discussion

RESPONSE: We adopted the ‘street level bureaucracy’ theory as a lens to understand the double layer of discretion exercised by local authority commissioners (in commissioning) and providers (in delivering) NHS Health Checks ‘on the ground’. For us, this theory provided a useful means of understanding the Health Check as a single case representing a wider phenomenon understood to affect other policies and programmes, where variation in delivery and divergence from policy intentions is apparent. Our more detailed findings can be understood as explanations for how this variation comes about in the case of the Health Check in particular.

8. COMMENT: Table 2 – should this include something on equity?

RESPONSE: Table 2 summarises the recommendations that we have developed on the basis of our findings. Our review identified a lack of data in relation to equity in the provision of advice, follow-up and referral to onward services – we have addressed this here in our recommendation relating to data monitoring (third line in the table), pointing out that such data would permit an assessment of any disparities in this provision.

References

1 Price T, Brennan N, Wong G, Withers L, Cleland J, Wanner A, et al. Remediation programmes for practising doctors to restore patient safety: the RESTORE realist review. Health Serv Deliv Res 2021;9(11)

2 Carrieri D, Pearson M, Mattick K, Papoutsi C, Briscoe S, Wong G & Jackson M. Interventions to minimise doctors' mental ill-health and its impacts on the workforce and patient care: the Care Under Pressure realist review. Health Serv Deliv Res 2020;8(19)

VERSION 2 – REVIEW

REVIEWER	Bonner, Carissa Western Sydney University School of Medicine
REVIEW RETURNED	06-Sep-2022
GENERAL COMMENTS	Although many of my comments remain unaddressed the authors have done a reasonable job of explaining the reasons behind the disconnect between the narrative results section and the data, with more details on the methods. These points should be included in the manuscript so other readers can understand this.

	Can you please include this explanation of your approach in the methods section of the paper so others are aware of the reasons behind this discrepancy: "Following the realist approach, our findings are an interpretation of a constellation of data extracted from the documents, built not only on direct literal quotes, but on silences and contrasts observed across multiple data points extracted from multiple documents. For example, our finding relating to a potential focus on clinical activity and high volume delivery of Health Checks is based on our interpretation of data that describe delivery models with this focus, as well as others that adopt a different stance. Taken together, we have surmised this is a reasonable interpretation of what is going on in some local areas, but there is no single piece of data explicitly demonstrating this." Can you acknowledge in your limitations section that other authors or readers with different backgrounds may draw different conclusions from your data given the method is described as follows in your response: "By providing the tables in Supplementary File 4 (and the corresponding CMOC information in the study table in Supplementary File 3) we have aimed for a level of transparency that permits access to the relevant data, such that interested readers could make their own judgement on the plausibility of our claims. "
--	--

VERSION 2 – AUTHOR RESPONSE

From Reviewer 2: *Although many of my comments remain unaddressed the authors have done a reasonable job of explaining the reasons behind the disconnect between the narrative results section and the data, with more details on the methods. These points should be included in the manuscript so other readers can understand this.*

Can you please include this explanation of your approach in the methods section of the paper so others are aware of the reasons behind this discrepancy: "Following the realist approach, our findings are an interpretation of a constellation of data extracted from the documents, built not only on direct literal quotes, but on silences and contrasts observed across multiple data points extracted from multiple documents. For example, our finding relating to a potential focus on clinical activity and high volume delivery of Health Checks is based on our interpretation of data that describe delivery models with this focus, as well as others that adopt a different stance. Taken together, we have surmised this is a reasonable interpretation of what is going on in some local areas, but there is no single piece of data explicitly demonstrating this."

Response: Thank you for this suggestion. We are keen that readers understand our application of realist analysis in this review, and so we have taken this on board. We have therefore added the suggested text describing how we worked with data within and across documents to build our interpretations (new text added on page 6). We also note that our protocol paper (also published in BMJ Open) provides readers with much more detail on our approach to analysis in this review.

Can you acknowledge in your limitations section that other authors or readers with different backgrounds may draw different conclusions from your data given the method is described as follows

in your response: "By providing the tables in Supplementary File 4 (and the corresponding CMOC information in the study table in Supplementary File 3) we have aimed for a level of transparency that permits access to the relevant data, such that interested readers could make their own judgement on the plausibility of our claims. "

Response: Thank you for this comment. We agree this is an inherent feature of any interpretive work, and have added a more explicit statement to this effect to our limitations section (on page 22).